# Coresets for Archetypal Analysis

**Sebastian Mair**
Leuphana University, Germany
`mair@leuphana.de`

**Ulf Brefeld**
Leuphana University, Germany
`brefeld@leuphana.de`

## Abstract

Archetypal analysis represents instances as linear mixtures of prototypes (the archetypes) that lie on the boundary of the convex hull of the data. Archetypes are thus often better interpretable than factors computed by other matrix factorization techniques. However, the interpretability comes with high computational cost due to additional convexity-preserving constraints. In this paper, we propose efficient coresets for archetypal analysis. Theoretical guarantees are derived by showing that quantization errors of $k$-means upper bound archetypal analysis; the computation of a provable absolute-coreset can be performed in only two passes over the data. Empirically, we show that the coresets lead to improved performance on several data sets.

## 1 Introduction

Archetypal analysis (Cutler and Breiman, 1994) is an unsupervised learning method that represents every data point as a convex combination of prototypes, the so-called archetypes. Every data point is represented as a convex mixture of (a subset of) archetypes and, due to the convexity, these mixtures are often interpreted probabilistically.

A key property of archetypal analysis is that the archetypes are themselves convex mixtures of data points. Consequently, archetypes lie on the boundary of the convex hull of the data. Hence, archetypal analysis approximates the convex hull with a given number of vertices. It follows that this approximation is equivalent to a matrix factorization of the design matrix. Due to the convexity constraints, archetypal-based factorizations are not only better interpretable but unfortunately also much more expensive than regular matrix factorization techniques, which hinders usage at even moderate scales.

Several approaches have been proposed to remedy the edacious nature of archetypal analysis, proposing, e.g., efficient active-set quadratic programming (Chen et al., 2014), projected gradients (Mørup and Hansen, 2012), or Frank-Wolfe techniques (Bauckhage et al., 2015) for optimization. Approximate solutions compute archetypes on a precomputed subset of the data, e.g., (Mair et al., 2017). Although these approaches are useful contributions, they do not mitigate the inherent complexity of archetypal analysis nor provide theoretical guarantees on the quality of the approximation.

A theoretically sound alternative is offered by coresets. Coresets compactly represent large data sets by weighted subsets on which models perform provably competitive compared to operations on all data. Coresets have successfully been leveraged to very different methods, including $k$-means (Lucic et al., 2016; Bachem et al., 2018), support vector machines (Tsang et al., 2005), logistic regression (Munteanu et al., 2018), and Bayesian inference (Huggins et al., 2016; Campbell and Broderick, 2018). The idea is as follows: A small subset of the data is selected (in linear time) according to a strategy such that the subset approximates the original data very well. A learning algorithm will then perform similarly on the original data and the subset, but training on the subset is much more efficient. In this paper, we present coresets for archetypal analysis.

The key contributions of this paper are as follows: (i) We show that the objective function of $k$-means upper bounds the objective of archetypal analysis and show that every coreset for $k$-means is also a

coreset for archetypal analysis, (ii) we propose a simple and efficient sampling strategy to compute a coreset with only two passes over the data, so that (iii) a weak $\varepsilon$-absolute-coreset is guaranteed to be obtained after sampling sufficiently many points where (iv) the error bound does not depend on the query itself. Finally, (v) we provide empirical results on various data sets to support the theoretical derivation.

## 2 Preliminaries

### 2.1 Archetypal Analysis

Let $\mathcal{X} = \{\mathbf{x}_1, \ldots, \mathbf{x}_n\}_{i=1}^n$ be a data set consisting of $n \in \mathbb{N}$ $d$-dimensional data points, $\mathbf{X} \in \mathbb{R}^{n \times d}$ be the design matrix and $k \in \mathbb{N}$ be the latent dimensionality. In archetypal analysis (Cutler and Breiman, 1994), every data point $\mathbf{x}_i$ is represented as a convex combination of $k$ archetypes $\mathbf{z}_1, \ldots, \mathbf{z}_k$, i.e.,

$$\mathbf{x}_i = \mathbf{Z}^\top \mathbf{a}_i, \quad \sum_{j=1}^d (\mathbf{a}_i)_j = 1, \quad (\mathbf{a}_i)_j \geq 0,$$

where $\mathbf{a}_i \in \mathbb{R}^k$ is the weight vector of the $i$th data point and $\mathbf{Z} \in \mathbb{R}^{k \times d}$ is the matrix of stacked archetypes. The archetypes $\mathbf{z}_j$ $(j = 1, \ldots, k)$ themselves are also represented as convex combinations of data points, i.e.,

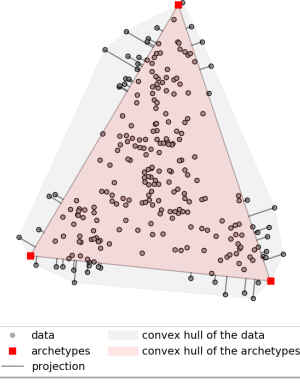

Figure 1: Archetypal analysis in two dimensions with $k = 3$ archetypes.

$$\mathbf{z}_j = \mathbf{X}^\top \mathbf{b}_j, \quad \sum_{i=1}^n (\mathbf{b}_j)_i = 1, \quad (\mathbf{b}_j)_i \geq 0,$$

where $\mathbf{b}_j \in \mathbb{R}^n$ is the weight vector of the $j$th archetype. Let $\mathbf{A} \in \mathbb{R}^{n \times k}$ and $\mathbf{B} \in \mathbb{R}^{k \times n}$ be the matrices consisting of the weights $\mathbf{a}_i$ $(i = 1, \ldots, n)$ and $\mathbf{b}_j$ $(j = 1, \ldots, k)$. Then, archetypal analysis yields a factorization of the design matrix as follows

$$\mathbf{X} = \mathbf{A}\mathbf{B}\mathbf{X} = \mathbf{A}\mathbf{Z}, \tag{1}$$

where $\mathbf{Z} = \mathbf{B}\mathbf{X} \in \mathbb{R}^{k \times d}$ is the matrix of archetypes. Due to the convexity constraints, the weight matrices $\mathbf{A}$ and $\mathbf{B}$ are row-stochastic. By minimizing the residual sum of squares (RSS), given by

$$\mathrm{RSS}(k) = \|\mathbf{X} - \mathbf{A}\mathbf{B}\mathbf{X}\|_F^2, \tag{2}$$

the optimal weight matrices $\mathbf{A}$ and $\mathbf{B}$ can be found. The objective function can be rewritten as a sum of projections of the data points to the archetype-induced convex hull as follows

$$\|\mathbf{X} - \mathbf{A}\mathbf{Z}\|_F^2 = \sum_{\mathbf{x} \in \mathcal{X}} \min_{\mathbf{q} \in \mathrm{conv}(\{\mathbf{z}_1, \ldots, \mathbf{z}_k\})} \|\mathbf{x} - \mathbf{q}\|_2^2,$$

where $\mathrm{conv}(S)$ refers to the convex hull of a set $S$. The minimization over points on the convex hull of the data renders the optimization infeasible for real sized problems. Although feasible optimization strategies (Chen et al., 2014; Bauckhage et al., 2015; Mørup and Hansen, 2012) and approximations (Mair et al., 2017; Mei et al., 2018) have been proposed, they all suffer from large dimensionalities and/or sample sizes.

### 2.2 Coresets

Let $\mathcal{X}$ be a data set of $n$ points in $d$ dimensions. Consider a learning problem with an objective function of the form $\phi_{\mathcal{X}}(Q) = \sum_{\mathbf{x} \in \mathcal{X}} d(\mathbf{x}, Q)^2$. The goal is to learn the so-called query $Q \subset \mathbb{R}^d$, with $|Q| = k$, and $d(\mathbf{x}, Q)^2$ is the minimal squared distance from a data point $\mathbf{x}$ to the query $Q$. For example, in $k$-means clustering, $Q$ refers to the set of cluster centers and $d(\mathbf{x}, Q)^2 = \min_{\mathbf{q} \in Q} \|\mathbf{x} - \mathbf{q}\|_2^2$. The objective function is then given by

$$\phi_{\mathcal{X}}(Q) = \sum_{\mathbf{x} \in \mathcal{X}} d(\mathbf{x}, Q)^2 = \sum_{\mathbf{x} \in \mathcal{X}} \min_{\mathbf{q} \in Q} \|\mathbf{x} - \mathbf{q}\|_2^2.$$

---
**Algorithm 1** Lightweight coreset construction for $k$-means (Bachem et al., 2018)
---
**Input:** Set of data points $\mathcal{X}$, coreset size $m$
**Output:** Coreset $\mathcal{C}$
$\mu \leftarrow$ mean of $\mathcal{X}$
**for** $x \in \mathcal{X}$ **do**
$\quad q(x) = \frac{1}{2}\frac{1}{|\mathcal{X}|} + \frac{1}{2}\frac{d(x,\mu)^2}{\sum_{x'} d(x',\mu)^2}$
**end for**
$\mathcal{C} \leftarrow$ sample $m$ points from $\mathcal{X}$ where each point has weight $\frac{1}{m \cdot q(x)}$ and is sampled with prob. $q(x)$

---

A coreset is a possibly weighted subset $\mathcal{C}$ of the full data set $\mathcal{X}$ with cardinality $m \ll n$, which performs provably competitive with respect to the performance on $\mathcal{X}$. Using non-negative weights $w_i \geq 0$ on the data points, the objective becomes $\phi_{\mathcal{X}}(Q) = \sum_{\mathbf{x} \in \mathcal{X}} w_i \cdot d(\mathbf{x}, Q)^2$. The standard definition of a coreset is as follows.

**Definition 1.** *Let $\varepsilon > 0$ and $k \in \mathbb{N}$. A (weighted) set $\mathcal{C}$ is a $(\varepsilon, k)$-coreset of the data $\mathcal{X}$ if for any $Q \subset \mathbb{R}^d$ of cardinality at most $k$*

$$|\phi_{\mathcal{X}}(Q) - \phi_{\mathcal{C}}(Q)| \leq \varepsilon \phi_{\mathcal{X}}(Q). \tag{3}$$

The condition in the definition of a coreset is equivalent to $(1-\varepsilon)\phi_{\mathcal{X}}(Q) \leq \phi_{\mathcal{C}}(Q) \leq (1+\varepsilon)\phi_{\mathcal{X}}(Q)$. Hence, the performance of the query learned on the coreset is bounded from below and above by a $(1 \pm \varepsilon)$ multiplicative of the query evaluated on the full data set. Note that Definition 1 defines a *strong* coreset since the bound holds uniformly for all queries $Q$. If the condition in Equation (3) holds only for the optimal query, $\mathcal{C}$ is called a *weak* coreset.

Computing a coreset for $k$-means may require $k$ sequential passes over the data (Lucic et al., 2016). Bachem et al. (2018) introduce the notion of lightweight-coresets, which allow for an additional additive term on the right hand side of the bound in Equation (3) and show that the solution can be computed in only two passes over the data. The definition of lightweight-coresets is as follows.

**Definition 2.** *(Bachem et al., 2018) Let $\varepsilon > 0$, $k \in \mathbb{N}$ and $\mathcal{X} \subset \mathbb{R}^d$ be a set of points with mean $\mu \in \mathbb{R}^d$. The weighted set $\mathcal{C}$ is a $(\varepsilon, k)$-lightweight-coreset of the data $\mathcal{X}$ if for any $Q \subset \mathbb{R}^d$ of cardinality at most $k$*

$$|\phi_{\mathcal{X}}(Q) - \phi_{\mathcal{C}}(Q)| \leq \frac{\varepsilon}{2}\phi_{\mathcal{X}}(Q) + \underbrace{\frac{\varepsilon}{2}\phi_{\mathcal{X}}(\{\mu\})}_{\textit{additive term}}. \tag{4}$$

The lightweight-coreset for $k$-means is constructed via importance sampling, in order to guide the sampling procedure towards more influential points. The sampling distribution $q$ is a mixture of an uniform distibution and the squared distances to the mean, i.e.,

$$q(\mathbf{x}) = \frac{1}{2}\frac{1}{n} + \frac{1}{2}\frac{d(\mathbf{x},\mu)^2}{\sum_{i=1}^{n} d(\mathbf{x}_i, \mu)^2}. \tag{5}$$

The underlying idea is that points that lie far away from the mean $\mu$ have a larger impact on the objective function and should thus be sampled with higher probability. The procedure is shown in Algorithm 1. After sampling $m$ points, each point is weighted by $(m \cdot q(\mathbf{x}))^{-1}$ such that the sampling procedure yields an unbiased estimator of the quantization error:

$$\mathbb{E}_{\mathcal{C}}\left[\phi_{\mathcal{C}}(Q)\right] = \mathbb{E}_{\mathcal{C}}\left[\sum_{\mathbf{x} \in \mathcal{C}} \frac{1}{m \cdot q(\mathbf{x})} d(\mathbf{x}, Q)^2\right] = \mathbb{E}\left[\frac{1}{q(\mathbf{x})} d(\mathbf{x}, Q)^2\right] = \sum_{\mathbf{x} \in \mathcal{X}} q(\mathbf{x})\frac{d(\mathbf{x}, Q)^2}{q(\mathbf{x})} = \phi_{\mathcal{X}}(Q).$$

The following result ensures that after sampling sufficiently many points, a $(\varepsilon, k)$-lightweight-coreset is obtained.

**Theorem 1** (Bachem et al. (2018)). *Let $\varepsilon > 0, \delta > 0$ and $k \in \mathbb{N}$. Let $\mathcal{X}$ be a set of points in $\mathbb{R}^d$ and let $\mathcal{C}$ be the output of Algorithm 1 with a sample size $m$ of at least*

$$m \geq c\frac{dk\log k + \log\frac{1}{\delta}}{\varepsilon^2},$$

*where $c$ is an absolute constant. Then, with probability of at least $1 - \delta$, $\mathcal{C}$ is a $(\varepsilon, k)$-lightweight-coreset of $\mathcal{X}$.*

Bachem et al. (2018) argue that dropping $\frac{\varepsilon}{2}\phi_\mathcal{X}(Q)$ from Equation (4) is not possible for the problem of $k$-means. Assume, for example, that the cluster centers (query $Q$) are placed arbitrarily far away from other data. Equation (4) would show an arbitrary large difference on the left hand side but the error on the right hand side would be bounded by $\frac{\varepsilon}{2}\phi_\mathcal{X}(\{\mu(\mathcal{X})\})$. Hence, $\mathcal{C}$ cannot be a coreset because it does not hold uniformly for all queries.

While this observation applies to $k$-means, the situation is very different for archetypal analysis. We assume that the mean $\mu$ of $\mathcal{X}$ is actually *contained in the convex hull* of the query, i.e., $\mu \in \mathrm{conv}(Q)$. Hence, placing some points of the query far away from the data induces a larger convex hull and thus a *lower* projection error. In the remainder, we also argue that queries of practical interest always lie on the border of the convex hulls of either $\mathcal{X}$ or $\mathcal{C}$, and that the mean $\mu$ is always included.

## 3   Coreset Construction

In the case of archetypal analysis, the query $Q$ consists of the archetypes $\mathbf{z}_1, \ldots, \mathbf{z}_k$. The squared distance of a point $\mathbf{x}$ to the query $Q$ is given by the length of the projection of the point to the convex set $\mathrm{conv}(Q)$, i.e., $d(\mathbf{x}, Q)^2 = \min_{\mathbf{q}\in\mathrm{conv}(Q)} \|\mathbf{x}-\mathbf{q}\|_2^2$. Hence, the objective function can be rewritten in the following way:

$$\phi_\mathcal{X}(Q) = \sum_{\mathbf{x}\in\mathcal{X}} d(\mathbf{x}, Q)^2 = \sum_{\mathbf{x}\in\mathcal{X}} \min_{\mathbf{q}\in\mathrm{conv}(Q)} \|\mathbf{x}-\mathbf{q}\|_2^2.$$

In the remainder, $\phi_\mathcal{X}(Q)$ refers to the above objective of archetypal analysis.

Before introducing and analyzing the coreset construction for archetypal analysis, we show that for a point $\mathbf{x}$ the quantization error of $k$-means upper bounds the projection of $\mathbf{x}$ to the query (i.e., the archetypes $\mathbf{z}_1, \ldots, \mathbf{z}_k$) in archetypal analysis. As a consequence, every coreset that bounds the error of $k$-means must also bound the error of archetypal analysis and is thus also a coreset for archetypal analysis.

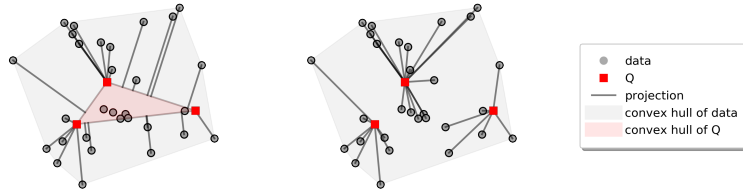

Figure 2: Illustration of Lemma 1. The projection of a point $\mathbf{x}$ to $\mathrm{conv}(Q)$ (left side) is smaller than or equal to the distance of $\mathbf{x}$ to the closest center $\mathbf{q} \in Q$ (right side).

**Lemma 1.** *Let $\mathbf{x} \in \mathbb{R}^d$ be a data point, $d(\cdot, \cdot)$ be a distance metric and $Q \subset \mathbb{R}^d$ be any set of $k \in \mathbb{N}$ points, then it holds that*

$$\min_{\mathbf{q}\in\mathrm{conv}(Q)} d(\mathbf{x}, \mathbf{q}) \leq \min_{\mathbf{q}\in Q} d(\mathbf{x}, \mathbf{q}).$$

*Proof.* First, note that $Q \subset \mathrm{conv}(Q)$. Assume that $\mathbf{q}' \in \mathrm{conv}(Q)$ minimizes $d(\mathbf{x}, \mathbf{q})$, then $\mathbf{q}'$ is either in $\mathrm{conv}(Q) \setminus Q$, resulting in a smaller distance than any other $\mathbf{q}'' \in Q$, or $\mathbf{q}'$ is in $Q$, yielding the same distance as $\min_{\mathbf{q}\in Q} d(\mathbf{x}, \mathbf{q})$. Hence, the distance of $\mathbf{x}$ to the convex set $\mathrm{conv}(Q)$ is smaller or equal to the distance to any $\mathbf{q} \in Q$. $\qquad\square$

A direct consequence of Lemma 1, which is depicted in Figure 2, is that for any choice of $Q$, the objective function of archetypal analysis is upper bounded by the objective of $k$-means, i.e.,

$$\sum_{\mathbf{x}\in\mathcal{X}} \min_{\mathbf{q}\in\mathrm{conv}(Q)} \|\mathbf{x}-\mathbf{q}\|_2^2 \leq \sum_{\mathbf{x}\in\mathcal{X}} \min_{\mathbf{q}\in Q} \|\mathbf{x}-\mathbf{q}\|_2^2.$$

Here, $Q$ in archetypal analysis refers to the archetypes $\mathbf{z}_1, \ldots, \mathbf{z}_k$ and in $k$-means, $Q$ refers to the set of centroids. Since a coreset bounds the error of a method on the entire set, and due to Lemma 1, any coreset for $k$-means is also a coreset for archetypal analysis.

---

**Algorithm 2** Coreset construction for Archetypal Analysis

---

**Input:** Set of data points $\mathcal{X}$, coreset size $m$
**Output:** Coreset $\mathcal{C}$
$\mu \leftarrow$ mean of $\mathcal{X}$
**for** $x \in \mathcal{X}$ **do**
$\quad q(x) = \frac{d(x,\mu)^2}{\sum_{x'} d(x',\mu)^2}$
**end for**
$\mathcal{C} \leftarrow$ sample $m$ points from $\mathcal{X}$ where each point has weight $\frac{1}{m \cdot q(x)}$ and is sampled with prob. $q(x)$

---

**Proposition 1.** *Every coreset for $k$-means is also a coreset for archetypal analysis.*

The proposition implies that the sampling strategy outlined in Algorithm 1 already yields a lightweight-coreset for our problem. However, we show in Section 3.1 that the term $\frac{\varepsilon}{2}\phi_{\mathcal{X}}(Q)$ can be dropped to obtain a weak $\varepsilon$-absolute-coreset for archetypal analysis, whose bound does not depend on the query $Q$.

**Definition 3.** *Let $\varepsilon > 0$ and $k \in \mathbb{N}$. A (weighted) set $\mathcal{C}$ is an $\varepsilon$-absolute-coreset of the data $\mathcal{X}$ if for any $Q \subset \mathbb{R}^d$ of cardinality at most $k$*

$$|\phi_{\mathcal{X}}(Q) - \phi_{\mathcal{C}}(Q)| \leq \varepsilon. \tag{6}$$

*The set $\mathcal{C}$ is called a weak $\varepsilon$-absolute-coreset if the bound holds only for specific queries $Q$.*

Archetypes are guaranteed to lie on the boundary of the convex hull of the data (Cutler and Breiman, 1994).[1] Thus, we are interested in points lying far from the mean $\mu$ of $\mathcal{X}$. Such points increase the convex hull of the archetypes and result in a smaller projections and hence in a lower value of the objective function. Following the idea of Bachem et al. (2018), we thus discard the uniform term in Equation (5) and propose the following sampling distribution

$$q(\mathbf{x}) = \frac{d(\mathbf{x}, \mu)^2}{\sum_{i=1}^{n} d(\mathbf{x}_i, \mu)^2}.$$

## 3.1 Analysis

We now provide a bound on the sample size $m$ to show that Algorithm 2 computes a provably competitive coreset.

**Theorem 2.** *Let $\varepsilon > 0, \delta > 0$ and $k \in \mathbb{N}$. Let $\mathcal{X}$ be a set of points in $\mathbb{R}^d$ with mean $\mu \in \mathbb{R}^d$ and let $\mathcal{C}$ be the output of Algorithm 2 with a sample size $m$ of at least*

$$m \geq c \frac{dk \log k + \log \frac{1}{\delta}}{\varepsilon^2},$$

*where $c$ is an absolute constant. Then, with probability of at least $1 - \delta$, the set $\mathcal{C}$ fulfills*

$$|\phi_{\mathcal{X}}(Q) - \phi_{\mathcal{C}}(Q)| \leq \varepsilon \phi_{\mathcal{X}}(\{\mu\}) \tag{7}$$

*for any query $Q \subset \mathbb{R}^d$ of cardinality at most $k$ satisfying $\mu \in \mathrm{conv}(Q)$.*

A proof of Theorem 2 can be found in the supplementary material. Note that the bound on the right hand side of Equation (7) is independent of the query $Q$ and corresponds to the scaled variance of the data. For a normalized data set, Algorithm 2 yields an $\varepsilon$-absolute-coreset as the following corollary shows.

**Corollary 1.** *Let $\varepsilon > 0, \delta > 0$, $k \in \mathbb{N}$, and $\mathcal{X}$ be a set of points in $\mathbb{R}^d$ with mean $\mu \in \mathbb{R}^d$. Denote by $\bar{\mathcal{X}}$ the standardized set of points with $\bar{\mathbf{x}}_i = (\mathbf{x}_i - \mu)/\phi_{\mathcal{X}}(\{\mu\})$. Let $\mathcal{C}$ be the output of Algorithm 2 on $\bar{\mathcal{X}}$ with a sample size $m$ of at least*

$$m \geq c \frac{dk \log k + \log \frac{1}{\delta}}{\varepsilon^2},$$

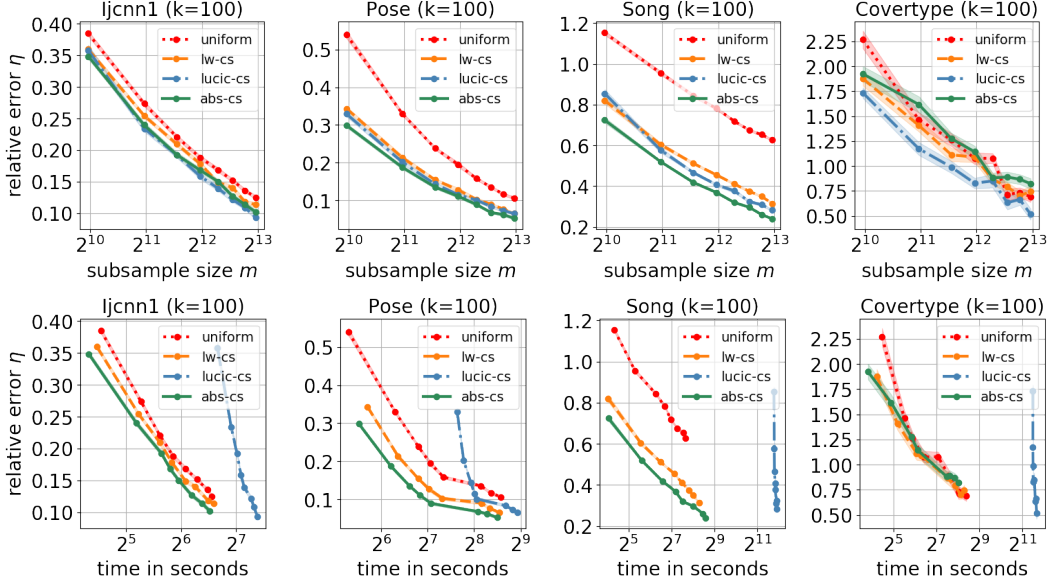

Figure 3: Relative error $\eta$ on the full data set as well as computation time in seconds (averages and standard errors of 50 independent runs).

*where $c$ is an absolute constant. Then, with probability of at least $1 - \delta$, $\mathcal{C}$ is an $\varepsilon$-absolute-coreset of $\bar{\mathcal{X}}$, i.e., it holds that*

$$|\phi_\mathcal{X}(Q) - \phi_\mathcal{C}(Q)| \leq \varepsilon$$

*for any query $Q \subset \mathbb{R}^d$ of cardinality at most $k$ satisfying $\mu \in \text{conv}(Q)$.*

Corollary 1 can be interpreted in the following way: As we decrease $\varepsilon$, the performance gap of archetypal analysis on the full (standardized) data set and archetypal analysis on the coreset closes for a query $Q$ satisfying $\mu \in \text{conv}(Q)$. One might ask whether this restriction on the choice of $Q$ is a drawback. The assumption within the various definitions of coresets that the bound has to hold for any choice of $Q$ is very strong. For the problem of $k$-means this makes sense as the centers could be anywhere in the space. However, for archetypal analysis, Cutler and Breiman (1994) show that the archetypes $\mathbf{z}_1, \ldots, \mathbf{z}_k$ lie on the boundary of the data, i.e., $\{\mathbf{z}_1, \ldots, \mathbf{z}_k\} \subset \partial\mathcal{C}$ for $k > 1$. Hence, any meaningful query $Q$ will be on the boundary of the coreset $\partial\mathcal{C}$ as well. Such a query will likely contain the mean $\mu$ of $\mathcal{X}$, because $\mathcal{C} \subset \mathcal{X}$ is sampled around $\mu$.

As the following theorem shows, the optimal solution $Q^\star_\mathcal{C}$ computed on the coreset $\mathcal{C}$ is indeed provably competitive to the solution $Q^\star_\mathcal{X}$ obtained on the full data set.

**Theorem 3.** *Let $\varepsilon > 0$ and $\mathcal{X}$ be a set of points in $\mathbb{R}^d$ with mean $\mu \in \mathbb{R}^d$. Denote by $Q^\star_\mathcal{X}$ the optimal solution on $\mathcal{X}$ and by $Q^\star_\mathcal{C}$ the optimal solution on $\mathcal{C}$. Then it holds that*

$$\phi_\mathcal{X}(Q^\star_\mathcal{C}) \leq \phi_\mathcal{X}(Q^\star_\mathcal{X}) + 2\varepsilon\phi_\mathcal{X}(\{\mu\}).$$

A proof of Theorem 3 is provided in the supplementary material.

### 3.2 Complexity Analysis

Algorithm 2 needs one full pass over the data set $\mathcal{X}$ of size $n$ in order to determine the mean $\mu$. Then, another pass is needed to compute the distance of each point $\mathbf{x}_i$ to the mean $\mu$ which is needed for the sampling distribution $q(\cdot)$. Hence, the complexity of Algorithm 2 scales in $\mathcal{O}(nd)$. In addition, since $q(\cdot)$ is a discrete distribution on $n$ objects, the space complexity is also in $\mathcal{O}(nd)$. The same arguments apply to the lightweight-coreset construction of (Bachem et al., 2018) as outlined in Algorithm 1.

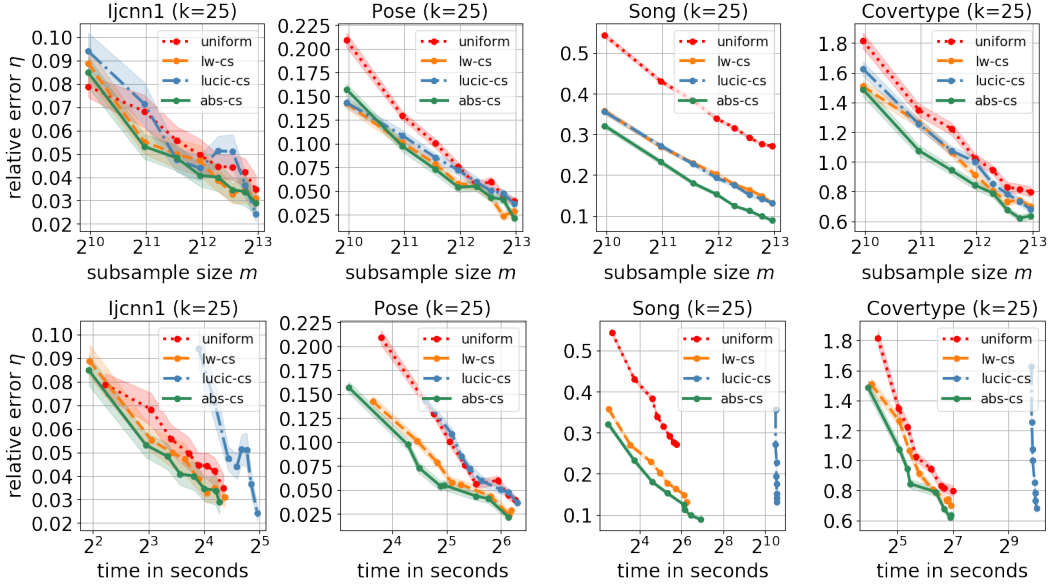

Figure 4: Relative error $\eta$ on the full data set as well as computation time in seconds (averages and standard errors of 50 independent runs).

## 4 Experiments

We now evaluate the coreset construction for archetypal analysis (**abs-cs**) and compare it to the performance of archetypal analysis on the full data set, an uniform sample (**uniform**), the lightweight-coresets for $k$-means (**lw-cs**, Bachem et al. (2018)), a state-of-the-art coreset construction for $k$-means (**lucic-cs**, Lucic et al. (2016)) as well as an approximate solution that learns archetypes on a precomputed subset (**frame**, Mair et al. (2017)).

We initialize the archetypes $\mathbf{z}_1, \ldots, \mathbf{z}_k$ using the furthest sum procedure (Mørup and Hansen, 2012). The termination criterion is reached when the relative error between iterations is less than $10^{-3}$. We measure the error in terms of the residual sum of squares (RSS) in Equation (2) and compute the relative error $\eta = (\mathrm{RSS}_{\mathrm{full}} - \mathrm{RSS}_{\mathrm{coreset}}) / \mathrm{RSS}_{\mathrm{full}}$ with respect to the performance on the full data set. We report on averages over 50 independent repetitions, error bars show standard error. The code is written in Python and all experiments run on an Intel Xeon machine with $28 \times 2.60\mathrm{GHz}$ and 256GB memory.[2]

We evaluate the algorithms on several data sets: **Ijcnn1** refers to data from the IJCNN 2001 neural network competition and has $n = 49,990$ instances in $d = 22$ dimensions.[3] We adopt the preprocessing from Chang and Lin (2001). **Pose** is a subset of the Human3.6M data set (Catalin Ionescu, 2011; Ionescu et al., 2014) and deals with 3D human pose estimation and is part of the ECCV 2018 Pose-Track Challenge.[4] It consists of $n = 35,832$ poses each of which is represented as 3D coordinates of 16 joints resulting in a 48-dimensional problem. **Song** is a subset of the Million Song Dataset (Bertin-Mahieux et al., 2011) which has $n = 515,345$ data points in $d = 90$ dimensions where the task is to predict the year of a song. **Covertype** (Blackard and Dean, 1999) contains $n = 581,012$ examples in $d = 54$ dimensions. The task is to predict the forest cover type from cartographic variables.

Figure 3 shows the results for $k = 100$ archetypes. In the top row, the relative error $\eta$ of each approach is evaluated on the full data set and illustrated for subsample sizes ranging from $m = 1,000$ to $m = 8,000$, depicted on the $x$-axis. Unsurprisingly, the relative error decreases with an increasing subsample size for all approaches. The uniform sampling strategy performs almost always worse than its peers. The coreset construction of Lucic et al. (2016) (*lucic-cs*) performs in some few cases on par

Table 1: Relative error in percent and speed up compared to the full data set for $k = 25$ archetypes (averages and standard errors of 50 independent runs).

| Data | Method | $m = 1000$ | | $m = 5000$ | |
|---|---|---|---|---|---|
| | | Relative Error | Speedup | Relative Error | Speedup |
| Covertype | uniform | 181.7%± 5.2% | 468× | 94.6%± 2.9% | 126× |
| | lw-cs | 150.8%± 4.1% | 553× | 80.6%± 2.5% | 119× |
| | lucic-cs | 162.7%± 4.8% | 10× | 85.4%± 2.6% | 9× |
| | abs-cs | 148.9%± 4.8% | 601× | 79.1%± 2.9% | 111× |
| Song | uniform | 54.4%± 0.6% | 430× | 31.7%± 0.4% | 78× |
| | lw-cs | 35.8%± 0.7% | 480× | 17.7%± 0.4% | 68× |
| | lucic-cs | 35.6%± 0.6% | 2× | 17.5%± 0.5% | 2× |
| | abs-cs | 32.1%± 0.6% | 486× | 12.4%± 0.3% | 39× |
| Pose | uniform | 20.9%± 0.8% | 9× | 5.6%± 0.4% | 3× |
| | lw-cs | 14.2%± 0.5% | 10× | 5.6%± 0.5% | 3× |
| | lucic-cs | 14.4%± 0.5% | 5× | 6.0%± 0.6% | 3× |
| | abs-cs | 15.7%± 0.6% | 14× | 5.5%± 0.5% | 4× |
| Ijccn1 | uniform | 7.9%± 0.5% | 17× | 4.5%± 0.5% | 5× |
| | lw-cs | 8.9%± 0.7% | 21× | 3.9%± 0.5% | 5× |
| | lucic-cs | 9.4%± 0.8% | 5× | 5.1%± 0.6% | 3× |
| | abs-cs | 8.5%± 0.6% | 21× | 4.0%± 0.6% | 6× |

with our proposed approach (*abs-cs*), see for example Ijcnn1. In most other cases, the proposed coreset construction yields the best results and outperforms its competitors, especially on the Song data.

The bottom row in Figure 3 also depicts the relative error $\eta$, however, with respect to the average runtime of a single run. Theoretically, the lightweight-coreset (*lw-cs*) as well as the proposed coreset construction realize complexities in $\mathcal{O}(nd)$. In practice, however, the proposed approach yields consistently lower relative errors in shorter time. We credit this finding to a better selection of coreset points resulting in a faster convergence of archetypal analysis. The method of Lucic et al. (2016) (*lucic-cs*) is consistently the slowest as it requires $k$ passes over the data for constructing the coreset.

Figure 4 shows the same as evaluation scenario as Figure 3 but for only $k = 25$ archetypes. Once again, our proposed coreset construction either outperforms its peers or performs on par with the lightweight-coreset construction of Bachem et al. (2018) while being more efficient. Table 1 summarizes the achieved speed-ups. On Covertype, for example, the computation of 25 archetypes with *abs-cs* and $m = 1,000$ is 601 times faster than computing the archetypes on the full data set. Although, the error is 148.9% higher than the error using archetypes learned on the full data set, all other competitors are consistently outperformed. Increasing the size of the coreset to $m = 5,000$ yields a much lower relative error of 79.1% while still being 111 times faster to compute. Similar results with less speed-up but also much less relative errors are obtained for the other data sets.

The remaining competitor *frame* (Mair et al., 2017) precomputes all data points lying on the boundary of the convex hull of the data set (the frame). We were not able to compute the frame within a reasonable amount of time for Covertype and Song.[5] On Pose, every data point lies on the boundary, hence the performance is identical to the performance on all data. For Ijcnn1, the number of points on the frame is about $0.57n$ and the relative error $\eta$ is about 0.03 for $k = 100$ archetypes. While this error is much lower, the subset size is also much larger. In addition, $m$ is not chosen but implicitly given by the data set.

## 5   Conclusion

We introduced coresets for archetypal analysis. The derivation was grounded on the observation that the quantization error of $k$-means serves as an upper bound on the projection error of archetypal analysis; hence, every coreset for $k$-means is also a coreset for archetypal analysis. We devised an algorithm based on importance sampling that computes a coreset in linear time with only two passes

over the data. A theoretical analysis showed that the proposed coreset performed competitive for a sufficiently large sample size. The theoretical results are supported by empiricism. The proposed algorithm outperformed its competitors on various data sets in terms of relative error and computation time. For some setups, we observed improved run-times. In sum, our contribution rendered archetypal analysis feasible for state-of-the-art-sized data sets.

## Footnotes

[1]Given that there is more than only a single archetype.

[2] https://github.com/smair/archetypalanalysis-coreset

[3] https://www.csie.ntu.edu.tw/~cjlin/libsvmtools/datasets/

[4] http://vision.imar.ro/human3.6m/challenge_open.php

[5]Computations take about 2,000 and 4,000 hours for Covertype and Song, respectively.

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
