[Supplementary Material]

# Supplementary Material of the Paper: Coresets for Archetypal Analysis

## A  Preliminaries

Before we proof Theorem 2, we introduce the concept of pseudo-dimension (Haussler, 1992; Li et al., 2001) and a result from Li et al. (2001).

**Definition 4.** *(Haussler, 1992; Li et al., 2001) Fix a countably infinite domain $\mathcal{X}$. The pseudo-dimension of a set $\mathcal{F}$ of functions from $\mathcal{X}$ to $[0, 1]$, denoted by $\mathrm{Pdim}(\mathcal{F})$, is the largest $d'$ such there is a sequence $x_1, \ldots, x_{d'}$ of domain elements from $\mathcal{X}$ and a sequence $r_1, \ldots, r_{d'}$ of reals such that for each $b_1, \ldots, b_{d'} \in \{above, below\}$, there is an $f \in \mathcal{F}$ such that for all $i = 1, \ldots, d'$ we have $f(x_i) \geq r_i \iff b_i = above$.*

**Theorem 4.** *(Li et al., 2001) Let $\alpha > 0$, $\nu > 0$ and $\delta > 0$. Fix a countably infinite domain $\mathcal{X}$ and let $P$ be any probability distribution over $\mathcal{X}$. Let $\mathcal{F}$ be a set of functions from $\mathcal{X}$ to $[0, 1]$ with $\mathrm{Pdim}(\mathcal{F}) = d'$. Denote by $\mathcal{C}$ a sample of $m$ points from $\mathcal{X}$ independently drawn according to $P$ with*

$$m \geq \frac{c}{\alpha^2 \nu} \left( d' \log \frac{1}{\nu} + \log \frac{1}{\delta} \right),$$

*where $c$ is an absolute constant. Then, it holds with probability of at least $1 - \delta$ that*

$$\mathrm{d}_\nu \left( \mathbb{E}_P[f], \frac{1}{|\mathcal{C}|} \sum_{x \in \mathcal{C}} f(x) \right) \leq \alpha \quad \forall f \in \mathcal{F},$$

*where $\mathrm{d}_\nu(a, b) = \frac{|a - b|}{a + b + \nu}$. Over all choices of $\mathcal{F}$ with $\mathrm{Pdim}(\mathcal{F}) = d$, this bound on $m$ is tight.*

## B  Proof of Theorem 2

The proof relies on bounds of the projection of a point $\mathbf{x}$ to the convex hull of the query $Q$.

**Lemma 2.** *Let $\mathcal{X}$ be a set of points in $\mathbb{R}^d$ with mean $\mu$. For all $\mathbf{x} \in \mathcal{X}$ and $Q \subset \mathbb{R}^d$ satisfying $\mu \in \mathrm{conv}(Q)$, it holds that*

$$d(\mathbf{x}, Q)^2 \leq 2d(\mathbf{x}, \mu)^2.$$

*Proof.* By the triangle inequality and since $(|a| + |b|)^2 \leq 2a^2 + 2b^2$ we have for any $\mathbf{x}$ and $Q$ that

$$d(\mathbf{x}, Q)^2 \leq (d(\mathbf{x}, \mu) + d(\mu, Q))^2 \leq 2d(\mathbf{x}, \mu)^2 + 2d(\mu, Q)^2.$$

Since $\mu \in \mathrm{conv}(Q)$, the distance of $\mu$ to the query $Q$ is zero, i.e., $d(\mu, Q)^2 = 0$ yielding our claim:

$$d(\mathbf{x}, Q)^2 \leq 2d(\mathbf{x}, \mu)^2.$$

$\square$

Theorem 2 can now be shown as follows.

**Theorem 2.** *Let $\varepsilon > 0, \delta > 0$ and $k \in \mathbb{N}$. Let $\mathcal{X}$ be a set of points in $\mathbb{R}^d$ with mean $\mu \in \mathbb{R}^d$ and let $\mathcal{C}$ be the output of Algorithm 2 with a sample size $m$ of at least*

$$m \geq c \frac{dk \log k + \log \frac{1}{\delta}}{\varepsilon^2},$$

*where $c$ is an absolute constant. Then, with probability of at least $1 - \delta$, the set $\mathcal{C}$ fulfills*

$$|\phi_\mathcal{X}(Q) - \phi_\mathcal{C}(Q)| \leq \varepsilon \phi_\mathcal{X}(\{\mu\}) \tag{7}$$

*for any query $Q \subset \mathbb{R}^d$ of cardinality at most $k$ satisfying $\mu \in \mathrm{conv}(Q)$.*

*Proof.* Let $\mu$ be the mean of $\mathcal{X}$. Consider the function

$$g_Q(x) = \frac{d(x, Q)^2}{2d(x, \mu)^2}.$$

Due to the non-negativity of the distances as well as Lemma 2 we know that $g_Q(x) \in [0, 1]$ for any $x \in \mathcal{X}$ and $Q \subset \mathbb{R}^d$ satisfying $\mu \in \text{conv}(Q)$. Then, it holds that

$$\phi_{\mathcal{X}}(Q) = \sum_{x \in \mathcal{X}} d(x, Q)^2 = \sum_{x \in \mathcal{X}} \frac{2d(x, \mu)^2 \phi_{\mathcal{X}}(\{\mu\})}{2d(x, \mu)^2 \phi_{\mathcal{X}}(\{\mu\})} d(x, Q)^2$$

$$= 2\phi_{\mathcal{X}}(\{\mu\}) \sum_{x \in \mathcal{X}} \frac{d(x, \mu)^2}{\phi_{\mathcal{X}}(\{\mu\})} \frac{d(x, Q)^2}{2d(x, \mu)^2} = 2\phi_{\mathcal{X}}(\{\mu\}) \sum_{x \in \mathcal{X}} q(x) g_Q(x) = 2\phi_{\mathcal{X}}(\{\mu\}) \mathbb{E}_q [g_Q(x)] .$$

Following the discussion in Bachem et al. (2017) and since every coreset for $k$-means is also a coreset for archetypal analysis (Proposition 1 in the paper), we use the result $\text{Pdim}(\mathcal{G}) \in \mathcal{O}(dk \log k)$. Hence, we can choose $d' = \frac{1}{\log 2} dk \log k$. Let $\alpha = \frac{\varepsilon}{6}$, $\nu = \frac{1}{2}$, $c'$ be an absolute constant and $c = 72c'$. By using

$$m \geq \frac{c'}{\alpha^2 \nu} \left( d' \log \frac{1}{\nu} + \log \frac{1}{\delta} \right) = \frac{72c'}{\varepsilon^2} \left( d' \log 2 + \log \frac{1}{\delta} \right) = c \frac{dk \log k + \log \frac{1}{\delta}}{\varepsilon^2},$$

Theorem 4 implies that with probability of at least $1 - \delta$

$$\text{d}_\nu \left( \mathbb{E}_q[g_Q(x)], \frac{1}{|\mathcal{C}|} \sum_{x \in \mathcal{C}} g_Q(x) \right) \leq \frac{\varepsilon}{6}$$

uniformly for all sets $Q$ of cardinality at most $k$ including $\mu$ in their convex hull. Since both arguments of $\text{d}_\nu$ are in $[0, 1]$, the denominator of $\text{d}_\nu$ is bounded by 3. Hence, we have

$$\left| \mathbb{E}_q[g_Q(x)] - \frac{1}{|\mathcal{C}|} \sum_{x \in \mathcal{C}} g_Q(x) \right| \leq \frac{\varepsilon}{2}.$$

We now multiply both sides by $2\phi_{\mathcal{X}}(\{\mu\})$ yielding

$$\left| 2\phi_{\mathcal{X}}(\{\mu\}) \mathbb{E}_q[g_Q(x)] - \frac{2\phi_{\mathcal{X}}(\{\mu\})}{|\mathcal{C}|} \sum_{x \in \mathcal{C}} g_Q(x) \right| \leq \varepsilon \phi_{\mathcal{X}}(\{\mu\}).$$

The first part is equal to $\phi_{\mathcal{X}}(Q)$ and the second part can be rewritten as follows:

$$\frac{2\phi_{\mathcal{X}}(\{\mu\})}{|\mathcal{C}|} \sum_{x \in \mathcal{C}} g_Q(x) = \frac{2\phi_{\mathcal{X}}(\{\mu\})}{|\mathcal{C}|} \sum_{x \in \mathcal{C}} \frac{d(x, Q)^2}{2d(x, \mu)^2} = \sum_{x \in \mathcal{C}} \frac{\phi_{\mathcal{X}}(\{\mu\})}{|\mathcal{C}| d(x, \mu)^2} d(x, Q)^2$$

$$= \sum_{x \in \mathcal{C}} \frac{1}{|\mathcal{C}| \frac{d(x, \mu)^2}{\phi_{\mathcal{X}}(\{\mu\})}} d(x, Q)^2 = \sum_{x \in \mathcal{C}} \frac{1}{|\mathcal{C}| q(x)} d(x, Q)^2 = \phi_{\mathcal{C}}(Q).$$

Finally, we obtain our claim:

$$|\phi_{\mathcal{X}}(Q) - \phi_{\mathcal{C}}(Q)| \leq \varepsilon \phi_{\mathcal{X}}(\{\mu\}).$$

$\square$

## C   Proof of Theorem 3

**Theorem 3.** *Let $\varepsilon > 0$ and $\mathcal{X}$ be a set of points in $\mathbb{R}^d$ with mean $\mu \in \mathbb{R}^d$. Denote by $Q_{\mathcal{X}}^\star$ the optimal solution on $\mathcal{X}$ and by $Q_{\mathcal{C}}^\star$ the optimal solution on $\mathcal{C}$. Then it holds that*
$$\phi_{\mathcal{X}}(Q_{\mathcal{C}}^\star) \leq \phi_{\mathcal{X}}(Q_{\mathcal{X}}^\star) + 2\varepsilon \phi_{\mathcal{X}}(\{\mu\}).$$

*Proof.* Since $Q_{\mathcal{C}}^\star$ is the optimal solution on $\mathcal{C}$ we know that
$$\phi_{\mathcal{C}}(Q_{\mathcal{C}}^\star) \leq \phi_{\mathcal{C}}(Q_{\mathcal{X}}^\star) \tag{8}$$
and by the property in Equation (7) we have that
$$\phi_{\mathcal{X}}(Q) - \varepsilon \phi_{\mathcal{X}}(\{\mu\}) \leq \phi_{\mathcal{C}}(Q) \leq \phi_{\mathcal{X}}(Q) + \varepsilon \phi_{\mathcal{X}}(\{\mu\}).$$
Inserting $Q_{\mathcal{C}}^\star$ and $Q_{\mathcal{X}}^\star$ yields
$$\phi_{\mathcal{X}}(Q_{\mathcal{C}}^\star) - \varepsilon \phi_{\mathcal{X}}(\{\mu\}) \leq \phi_{\mathcal{C}}(Q_{\mathcal{C}}^\star) \leq \phi_{\mathcal{X}}(Q_{\mathcal{C}}^\star) + \varepsilon \phi_{\mathcal{X}}(\{\mu\}), \tag{9}$$
$$\phi_{\mathcal{X}}(Q_{\mathcal{X}}^\star) - \varepsilon \phi_{\mathcal{X}}(\{\mu\}) \leq \phi_{\mathcal{C}}(Q_{\mathcal{X}}^\star) \leq \phi_{\mathcal{X}}(Q_{\mathcal{X}}^\star) + \varepsilon \phi_{\mathcal{X}}(\{\mu\}). \tag{10}$$
It follows that
$$\phi_{\mathcal{X}}(Q_{\mathcal{C}}^\star) - \varepsilon \phi_{\mathcal{X}}(\{\mu\}) \overset{(9)}{\leq} \phi_{\mathcal{C}}(Q_{\mathcal{C}}^\star) \overset{(8)}{\leq} \phi_{\mathcal{C}}(Q_{\mathcal{X}}^\star) \overset{(10)}{\leq} \phi_{\mathcal{X}}(Q_{\mathcal{X}}^\star) + \varepsilon \phi_{\mathcal{X}}(\{\mu\}).$$
Adding $\varepsilon \phi_{\mathcal{X}}(\{\mu\})$ to both sides yields the claim. $\square$

---
**Algorithm 3** Archetypal Analysis (Cutler and Breiman, 1994)
---
    **Input:** data matrix $\mathbf{X}$, number of archetypes $k$
    **Output:** weight matrices $\mathbf{A}$ and $\mathbf{B}$
    $\mathbf{Z} \leftarrow$ initialization of the archetypes, e.g., via FurthestSum (Mørup and Hansen, 2012)
    **while** not converged **do**
        **for** $i = 1, 2, \ldots, n$ **do**
            $\mathbf{a}_i = \underset{\|\mathbf{a}_i\|_1 = 1, \mathbf{a}_i \geq 0}{\arg\min} \|\mathbf{Z}^\top \mathbf{a}_i - \mathbf{x}_i\|_2^2$
        **end for**
        $\mathbf{Z} = (\mathbf{A}^\top \mathbf{A})^{-1} \mathbf{A}^\top \mathbf{X}$
        **for** $j = 1, 2, \ldots, k$ **do**
            $\mathbf{b}_j = \underset{\|\mathbf{b}_j\|_1 = 1, \mathbf{b}_j \geq 0}{\arg\min} \|\mathbf{X}^\top \mathbf{b}_j - \mathbf{z}_j\|_2^2$
        **end for**
        $\mathbf{Z} = \mathbf{B}\mathbf{X}$
    **end while**
---

## D   Weighted Archetypal Analysis

Algorithm 2 does not only produce a subset $\mathcal{C}$ of $\mathcal{X}$ but also corresponding weights $w_i > 0$ ($i = 1, \ldots, m$) of the sampled data points. These weights need to be incorporated into the learning procedure.

Eugster and Leisch (2011) propose a weighted archetypal analysis by moving the weights into the Frobenius norm in Equation (2) as follows

$$\sum_{i=1}^{n} w_i \|\mathbf{x}_i - \mathbf{Z}^T \mathbf{a}_i\|_2^2 = \sum_{i=1}^{n} w_i \sum_{j=1}^{d} ((\mathbf{x}_i)_j - (\mathbf{Z}^T \mathbf{a}_i)_j)^2 = \sum_{i=1}^{n} \sum_{j=1}^{d} ((\sqrt{w_i}\mathbf{x}_i)_j - (\sqrt{w_i}\mathbf{X}^T \mathbf{B}^T \mathbf{a}_i)_j)^2$$

$$= \sum_{i=1}^{n} \sum_{j=1}^{d} ((\tilde{\mathbf{x}}_i)_j - (\tilde{\mathbf{X}}^T \mathbf{B}^T \mathbf{a}_i)_j)^2,$$

where the $\tilde{\mathbf{x}}_i = \sqrt{w_i}\mathbf{x}_i$ ($i = 1, \ldots, n$) denote the transformed data points. Once the data has been transformed, a standard archetypal analysis can be computed. Note that, before application, the weight matrix $\mathbf{A}$ needs to be re-computed on the original data using the archetypes. Unfortunately, this approach has a major drawback: Since the archetypes $\mathbf{z}_j$ live on the boundary of $\mathrm{conv}(\mathcal{X})$, scaling $\mathcal{X}$ will change the support of the archetypes. This is however counterintuitive in our setting. While the placement of the archetypes may be influenced by the weights, their support should not be affected.

We thus propose an alternative way to include weights into vanilla archetypal analysis as proposed by Cutler and Breiman (1994) and outlined in Algorithm 3. Without loss of generality, assume that the weights were natural numbers. Hence, having a weight $w_i$ on data point $\mathbf{x}_i$ is equivalent to having the data point $w_i$ times within the data set. This causes $w_i$ times the projection $\|\mathbf{x}_i - \mathbf{Z}^T \mathbf{a}_i\|_2^2$. In addition, we have $w_i$ times the weight vector $\mathbf{a}_i$. Thus, the weight $w_i$ affects the intermediate update of the archetypes, which is $\mathbf{Z} = (\mathbf{A}^\top \mathbf{A})^{-1} \mathbf{A}^\top \mathbf{X}$ as outlined in Algorithm 3. Instead of adding a point $\mathbf{x}_i$ multiple ($w_i$) times, we can incorporate the weights via the modified update rule

$$\mathbf{Z} = (\tilde{\mathbf{A}}^\top \tilde{\mathbf{A}})^{-1} \tilde{\mathbf{A}}^\top \tilde{\mathbf{X}},$$

where $\tilde{\mathbf{A}} = \mathbf{W}\mathbf{A}$, $\tilde{\mathbf{X}} = \mathbf{W}\mathbf{X}$ and $\mathbf{W}$ is the diagonal matrix of rooted weights, i.e., $W_{ii} = \sqrt{w_i}$. Note that by this change, the location but not the support of the archetypes $\mathbf{z}_1, \ldots, \mathbf{z}_k$ is altered with respect to the weights. The generalization to real-valued weights $w_i > 0$ follows directly.