[Reviews · NeurIPS 2019]

Reviewer 1



This paper looks at the problem of archetypal analysis -- which effectively is a low-rank representation of the data that perhaps has more interpretablility. Instead of finding a low-rank subspace to represent the data, we try to represent each data point as a projection to a convex hull of k-points, where the k points themselves are convex combinations of the original data. The authors present a sampling based method to create coresets for this problem. The main intuition is that the objective function is close to (in fact upper bounded by) a k-means objective, just the "query set" has changed. Given the strong coreset guarantee, a restriction of the query set means that the existing guarantees carry over. They use a modification of an existing efficient sampling technique and obtain an additive guarantee. Overall, the connection is nice, though perhaps not very surprising to someone familiar with the kmeans coreset guarantees. The experiments do definite improvement over uniform sampling and some improvement over the existing "lightweight coreset". I wonder why the original kmeans coreset was not used in experiments-- the datasets are small enough that k passes is not too expensive. Few editorial comments: The dimension of B is stated wrongly, it should be k X n. Correspondingly that of A. I did not understand the comment that "removing \eps/2 \phi_X(Q) causes problem in clustering"-- what the authors probably mean is that we cannot really select a subset of points that satisfies the condition without this term on the RHS, for all centers Q. Also, the statement "In contrast to k-means, we assume that the mean .." is not clear to me. Either this is implied by the nature of the queries in this optimization problem (which is what I suspect), or it is an important assumption that must be stated clearly. Just noting that the RHS of the bound 7 implies that we get an additive, and not multiplicative approximation-- most coresets go for a multiplicative guarantee. I wonder if Proposition 1 is really accurate as stated? If we take the original multiplicative error definition of coreset, then I do not see an obvious argument, in spite of the lower bound in Lemma 1. It seems true if we defining coresets in terms of absolute error, which is what is used here (Definition 3). It will be nice to clarify/present a proof in supplementary. Overall, making the connection to the k-means objective is not very surprising. The fact that the lightweight k-means (and a slight modification) gives as additive error for coresets is neat.

Reviewer 2



Summary: Archetypal Analysis (AA) provides a framework in which a dataset is described as a convex combination of prototypes which represent vertices approximating the convex hull of the data. These archetypes allow for simplified interpretation but are difficult to infer for large datasets. In this regard, coresets can be used to compactly represent the dataset. In this paper, lightweight coresets for k-means are extended to AA and performance guarantees of the coreset C compared to the full dataset X and an upper bound on the sample size of C are derived and rigorously proved. A sampling procedure for the construction of the coreset in AA in linear time is provided. Experimental results on different dataset support the effectiveness of the novel approach compared to uniform sampling of the data to form a coreset and the underlying lightweight-coresets for k-means approach. The results on the relative error of the new approach compared to AA on the full dataset suggest room for improvement. Overview of the metrics: - Originality: The main contribution is the extension of lightweight coresets for k-means to AA, where theorems and algorithms are adjusted to the archetype setting. The problem is well motivated and the extension enriches methods on coresets. - Quality: Claims are well supported and proofs are provided. The experimental results illustrate the benefits of the method and possible weaknesses (large relative error compared to AA on full dataset) are reported. - Clarity: The submission is well written and easy to follow, the concept of coresets is well motivated and explained. While some more implementation details could be provided (source code is intended to be provided with camera-ready version), a re-implementation of the method appears feasible. - Significance: The submission provides a method to perform (approximate) AA on large datasets by making use of coresets and therefore might be potentially useful for a variety of applications. Detailed remarks/questions: 1. Algorithm 2 provides the coreset C and the query Q consists of the archetypes z_1, …, z_k which are initialised with the FurthestSum procedure. However, it is not quite clear to me how the archetype positions are updated after initialisation. Could the authors please comment on that? 2. The presented theorems provide guarantees for the objective functions phi on data X and coreset C for a query Q. Table 1 reporting the relative errors suggests that there might be a substantial deviation between coreset and full dataset archetypes. However, the interpretation of archetypes in a particular application is when AA proves particularly useful (as for example in [1] or [2]). Is the archetypal interpretation of identifying (more or less) stable prototypes whose convex combinations describe the data still applicable? 3. Practically, the number of archetypes k is of interest. In the presented framework, is there a way to perform model selection in order to identify an appropriate k? 4. The work in [3] might be worth to mention as a related approach. There, the edacious nature of AA is approached by learning latent representation of the dataset as a convex combination of (learnt) archetypes and can be viewed as a non-linear AA approach. [1] Shoval et al., Evolutionary Trade-Offs, Pareto Optimality, and the Geometry of Phenotype Space, Science 2012. [2] Hart et al., Inferring biological tasks using Pareto analysis of high-dimensional data, Nature Methods 2015. [3] Keller et al., Deep Archetypal Analysis, arxiv preprint 2019. ---------------------------------------------------------------------------------------------------------------------- I appreciate the authors’ response and the additional experimental results. I consider the plot of the coreset archetypes on a toy experiment insightful and it might be a relevant addition to the appendix. In my opinion, the submission constitutes a relevant contribution to archetypal analysis which makes it more feasible in real-world applications and provides some theoretical guarantees. Therefore, I raise my assessment to accept.

Reviewer 3



Archetypal analysis is a type of matrix factorization approach that is more interpretable but at the same time more computationally expensive. Finding coresets of this problem helps finding archetypes for large datasets more efficiently. The theoretical results presented in the paper are quite useful in that sense. One of the concerns I have with the theorems is that archetypal analysis finds extreme points and is therefore more sensitive to perturbation than k-means. Take for example samples in one dimension coming from a bimodal distribution. k-means with k=2 should find the two modes of the distribution, and should be quite stable if we take a subsample of the data. Archetypal analysis on the other hand should find the minimum and maximum values of the samples, and therefore is sensitive to subsampling. The results presented in the paper only establish bounds on the residuals of archetypal analysis for using full dataset versus the using coresets, i.e, \phi(C,Q_C) versus \phi(X,Q_X). However, the archetypes found by the coreset and the original dataset can be different, and it will be interesting to have theoretical properties for |Q_C - Q_X|. This can also be shown computationally in the experiment section. Can the authors please comment on this? In the experiments section, the uniform sampling works quite similar to coresets for covertype and ijcnn1 while it does not work well for pose and song. Can the authors comment on why this is so? Do the two former datasets have more uniformly distributed samples than the latter datasets? Since the author are reporting relative error, the y axis of the figures should start from 0. lines 104-114 seem a bit out of place. I am guessing that the authors are trying to establish the link between definition 4 and theorem 2 where the RHS does not depend on the eps/2... term. This should be clarified. in line 97-98, please state with respect to what the expectation is taken over, i.e., coresets

[Author Response · NeurIPS 2019]

We thank all reviewers for their careful reading and their valuable comments. In the following, we answer the main questions and comment on the points raised. **R1:** *[..] I wonder why the original kmeans coreset was not used in experiments [..]* The experimental analysis of Bachem et al. (2018) shows that the lightweight-coreset performs very similar to the one in Lucic et al. (2016) for $k$-means. As seen in the figure on the right, the performance of Lucic et al. (2016) (lucic-cs) is indeed very similar to the lightweight-coreset, even for AA. We now included this baseline in the paper.

**R1:** *The dimension of B is stated wrongly [..]* Thank you for pointing us to the typo in the dimensionalities of the matrices $\mathbf{A}$ and $\mathbf{B}$. We revised the manuscript accordingly. **R1:** *I did not understand the comment that "removing $\frac{\varepsilon}{2}\phi_X(Q)$ causes problem in clustering" [..] Also, the statement "In contrast to k-means, we assume that the mean .." is not clear to me.* Thank you for raising this issue. Reviewer 3 also pointed this out. We revised these paragraphs accordingly. **R2:** *[..] However, it is not quite clear to me how the archetype positions are updated after initialisation. [..]* After initialization, the matrices $\mathbf{A}$ and $\mathbf{B}$ ($\mathbf{Z} = \mathbf{BX}$) are optimized such that the residual sum of squares (RSS in Eq. (2)) is minimal. The standard procedure of the alternating optimization over $\mathbf{A}$ for fixed $\mathbf{B}$ and vice versa is also outlined in Algorithm 1 in the supplementary material.

Song (k=25)

**R2:** *[..] Table 1 reporting the relative errors suggests that there might be a substantial deviation between coreset and full dataset archetypes. [...]* Note that the "large" relative errors may be due to a too small coreset size $m$ for this data set and that we chose the same coreset sizes for all data sets. By increasing $m$, the relative error is expected to drop further. In practice, one does usually not choose $\varepsilon$ and $\delta$ and compute the correct $m$ but rather uses the largest $m$ suitable for the infrastructure at hand. Our theoretical results ensure that the errors are bounded and that the approach is better than a naive uniform subsample. **R2:** *[..] Is the archetypal interpretation of identifying (more or less) stable prototypes whose convex combinations describe the data still applicable?* **R3:** *[..] However, the archetypes found by the coreset and the original dataset can be different, and it will be interesting to have theoretical properties for $|Q_C - Q_X|$ [..]* The archetypes found by the coreset and the full data set are indeed different. Otherwise, the relative error was zero since we always report error on the full data set. Measuring $\|Q_C - Q_X\|_F$ as suggested is not trivial since the archetypes in the $\mathbf{Z}$ matrices might be permuted. Hence, we would have to rely on something like optimal transport on empirical distributions. However, even if we computed those, the errors would drop for increasing coreset sizes $m$. As $m$ approaches $n$, the archetypes on the coresets converge towards the archetypes on the full data set. It is also not clear how to measure interpretability. The reviewers are totally right by stating that AA is naturally more sensitive to points on the boundary of the data. This is one reason why we dropped the uniform part within the sampling distribution (compare Eq. (5) with line 148) to put more focus on points far away of the center of data. By increasing $m$, more of those points are discovered and the archetypes can be placed closer to the real boundary. However, the directions in which the archetypes lie should be approximatively preserved. We conducted a simple experiment on toy data with $n = 250$ points and sampled 10 coresets for each $m \in \{25, 50, 75, 125\}$. The archetypes learned on the coresets (blue) converge to the archetypes learned on all data (red). The larger the coreset the better the approximation.

**R2:** *Practically, the number of archetypes k is of interest. In the presented framework, is there a way to perform model selection in order to identify an appropriate k?* The standard way of model selection in AA is to compute the RSS for various values of $k$, then plot the error against $k$ and finally choosing the $k$ according to the elbow criterion. This can be also done on a coreset. **R2:** *The work in [3] might be worth to mention as a related approach. [..]* Thank you for pointing us to this related literature; the new version now includes a discussion in the related work section. **R3:** *Since the author are reporting relative error, the y axis of the figures should start from 0* For most of the plots it won't make a difference, but for covertype it would add a lot of white space and render differences in performance very small. **R3:** *in line 97-98, please state with respect to what the expectation is taken over* Thank you. We updated this part.

[Meta-Review · NeurIPS 2019]

This paper provides a core-set construction for the problem of Archetypal Analysis, in which a dataset must be described as a convex combination of prototypes which represent vertices approximating the convex hull of the data. Their core-set construction extends from that for the k-means problem. In addition to their approximation algorithm to perform archetypal analysis, which can run on large data sets, they provide theoretical analysis. The reviewers were unanimous in their vote to accept. Authors are encouraged to revise with respect to reviewer comments.